# Subthreshold micropulse laser versus standard laser for the treatment of central-involving diabetic macular oedema with central retinal thickness of <400μ: a cost-effectiveness analysis from the DIAMONDS trial

Hema Mistry ![ORCID],[1,2] Mandy Maredza,[1] Christina Campbell,[3] Noemi Lois,[4] On behalf of the DIAMONDS study group

For numbered affiliations see end of article.

**Correspondence to**
Dr Hema Mistry;
Hema.Mistry@warwick.ac.uk

## ABSTRACT

**Objectives** To estimate the economic costs, health-related quality-of-life outcomes and cost-effectiveness of subthreshold micropulse laser (SML) versus standard laser (SL) for the treatment of diabetic macular oedema (DMO) with central retinal thickness (CRT) of <400μ.

**Design** An economic evaluation was conducted within a pragmatic, multicentre, randomised clinical trial, DIAbetic Macular Oedema aNd Diode Subthreshold.

**Setting** 18 UK Hospital Eye Services.

**Participants** Adults with diabetes and centre involving DMO with CRT<400μ.

**Interventions** Participants (n=266) were randomised 1:1 to receive SML or SL.

**Methods** The base-case used an intention-to-treat approach conducted from a UK National Health Service (NHS) and personal social services (PSS) perspective. Costs (2019–2020 prices) were collected prospectively over the 2-year follow-up period. A bivariate regression of costs and quality-adjusted life-years (QALYs), with multiple imputation of missing data, was conducted to estimate the incremental cost per QALY gained and the incremental net monetary benefit of SML in comparison to SL. Sensitivity analyses explored uncertainty and heterogeneity in cost-effectiveness estimates.

**Results** One participant in the SL arm withdrew consent for data to be used; data from the remaining 265 participants were included in analyses. Mean (SE) NHS and PSS costs over 24 months were £735.09 (£111.85) in the SML arm vs £1099.70 (£195.40) in the SL arm (p=0.107). Mean (SE) QALY estimates were 1.493 (0.024) vs 1.485 (0.020), respectively (p=0.780), giving an insignificant difference of 0.008 QALYs. The probability SML is cost-effective at a threshold of £20 000 per QALY was 76%.

**Conclusions** There were no statistically significant differences in EQ-5D-5L scores or costs between SML and SL. Given these findings and the fact that SML does not burn the retina, unlike SL and has equivalent efficacy to SL, it may be preferred for the treatment of people with DMO with CRT<400μ.

### STRENGTHS AND LIMITATIONS OF THIS STUDY

⇒ Study is based on a high-quality randomised clinical trial.
⇒ First study to compare the cost-effectiveness of subthreshold micropulse laser with that of standard laser for the treatment of diabetic macular oedema.
⇒ The analysis was from a UK National Health Service (NHS) and personal social services perspective and did not consider costs such as productivity losses from a societal perspective.
⇒ Low rate of missing data for both costs and outcomes.
⇒ We used published costs of anti-vascular endothelial growth factor drugs, rather than NHS costs which incorporates confidential price discounts for these drugs.

**Trial registration numbers** ISRCTN17742985; NCT03690050.

## INTRODUCTION

Diabetic macular oedema (DMO) is a visual-threatening complication of diabetes, occurring in approximately 7% of people living with diabetes.[1] Given the high and continuously increasing prevalence of diabetes,[2] DMO is a frequent eye disease requiring treatment in ophthalmic clinics in the UK and worldwide. DMO can impose a significant social and economic burden on society, due to its high prevalence and associated costs. Very few studies have explored the economic burden of DMO and even fewer have reported on its cost-effectiveness. A cost-of-illness study using cohort data from US Medicare data reported just under 38% with DMO underwent laser photocoagulation and their 1-year

mean direct medical costs amounted to US$11 290, 31% higher than for those without DMO.[3] In a cohort study from South Korea, the mean 1-year medical costs were higher for people with DMO (US$6723) than those who had diabetes without retinopathy.[4] The estimated health-care and social care costs for DMO in England in 2010 were £92 million and £11.6 million, respectively, with £65.6 million of this being spent on hospital treatment and related costs.[5]

In DMO fluid accumulates in the centre of the retina, the macula, which is the area of the retina responsible for providing central vision.[6] The purpose of the treatment is to restore the anatomy of the macula by clearing up this fluid and restoring vision. Treatments include intraocular injections of antivascular endothelial growth factor (anti-VEGF) drugs or steroids and macular laser. The National Institute for Health and Care Excellence (NICE) recommends anti-VEGF therapy for people with more severe forms of DMO, with central retinal (macular) thickness (CRT) of 400µ or above, as measured in scans obtained using an imaging modality called optical coherence tomography (OCT).[7–9] For milder forms of DMO (CRT<400µ), NICE recommends macular laser. Intravitreal steroids are advised for people that do not respond to the above-mentioned therapies.

DIAbetic Macular Oedema aNd Diode Subthreshold micropulse laser (DIAMONDS) was a pragmatic, allocation-concealed, double-masked, multicentre, randomised, non-inferiority clinical trial which compared the clinical effectiveness and cost-effectiveness of subthreshold micropulse laser (SML) and standard laser (SL) for the treatment of people with DMO with CRT of <400µ.[10 11] DIAMONDS participants were randomised 1:1 to receive SML (577 nm) or SL (eg, using argon, frequency-doubled neodymium-doped yttrium aluminium garnet (Nd:YAG) 532 nm laser). Laser treatment could be repeated as needed, using the allocated laser at randomisation and rescue treatment with anti-VEGFs or steroids was allowed. The primary outcome was the mean change in best-corrected visual acuity (BCVA) in the study eye at 24 months. DIAMONDS found SML and SL to have equivalent clinical efficacy.[10 11] This finding is clinically important given the fact that SML, unlike SL, does not cause any functional or structural damage to the retina[12–14] and, thus, may be preferred by patients and doctors. Here, we present the detailed within-trial economic evaluation comparing costs and benefits of the two laser modalities, SML and SL. To our knowledge, no other trials have compared the cost-effectiveness of SML and SL for the treatment of DMO before.

## METHODS
### Patient and public involvement
As described previously[10 11]: 'At the very early stages of the DIAMONDS trial conception, a DIAMONDS Patient and Public Involvement (PPI) group was established with the help of the Northern Ireland branch of DIABETES

UK. The DIAMONDS PPI group comprised people living with diabetes and DMO, including a large group of members of the 'Diabetes Family' Facebook group. The DIAMONDS PPI group contributed to the trial design and the research question, including the selection of outcomes, preparation of patient related materials for the trial, recruitment strategies, interpretation of trial results and preparation of the plain English summary. They also have a major role in the dissemination and implementation of trial results.'[10 11]

The study is reported as per Consolidated Health Economic Evaluation Reporting Standards 2022 Statement.[15] As detailed in the Health Economics Analysis Plan,[16] we planned to conduct a within-trial analysis comparing the cost-effectiveness of SML with SL. The protocol for the DIAMONDS trial envisaged that economic modelling might be required if visual outcomes differed between arms.[17] The DIAMONDS protocol was designed to minimise visual loss in participants and, thus, repeating laser treatment or undertaking rescue treatment with intravitreal anti-VEGF drugs and/or steroids if criteria for rescue were met, were allowed in either arm of the trial. DIAMONDS recruited 266 participants, 116 (87%) in the SML arm and 115 (86%) in the SL arm had primary outcome data, fulfilling the requirements of the power calculation (113 participants with BCVA data at month 24 were required). DIAMONDS found SML and SL to be equivalent in terms of the primary outcome.[10 11] Hence, economic modelling was not required. Here, we report detailed economic costs, health-related quality of life (HRQoL) outcomes and cost-effectiveness of SML vs SL.

As described in detail previously in Lois et al[10] and Lois et al,[11] the methods (resource use, costs and outcomes) have been summarised here. The economic evaluation took the form of a cost–utility analysis, expressed in terms of cost per quality-adjusted life-year (QALY) gained. The study was conducted over the 24-month time horizon of the trial and adopted a National Health Service (NHS) and personal and social service (PSS) perspective. Costs and outcomes in the second year of follow-up were discounted at 3.5% in line with the NICE reference case.[18]

Resource use data were collected and reported on trial case record forms (CRFs) at scheduled 4-monthly clinic visits (4, 8, 12, 16, 20 and 24 months). Data were collected on the costs of laser treatment, both at the initial laser session and at subsequent ones if required, outpatient visits and intravitreal anti-VEGF and/or steroid treatment (costs of drugs and administration) if rescue was required. All costs were expressed in UK pounds sterling and valued in 2019–2020 prices. If costs were not in line, they were inflated to 2019–2020 prices using the NHS Cost Inflation Index.[10 19]

The costs of laser treatment included staff and equipment costs (capital and maintenance costs of laser machines). Unit costs for staff were obtained from the Unit Costs of Health and Social Care 2019 compendium[19] and were based on the time it took to for each procedure

to be undertaken. These times were recorded on CRFs and included: (1) the time taken to obtain fundus fluorescein angiography (FFA) and spectral domain optical coherence tomography (SD-OCTs) scans to guide laser treatment (if used) and (2) time taken by ophthalmologists to perform the laser procedure, including counselling the participant. Costs of laser machines were obtained from manufacturers. An annual equivalent cost of equipment was obtained by annuitising the capital costs of the item over its useful life span and applying a discount rate of 3.5% per annum.[10] A per-patient cost of equipment was estimated by assuming that the laser machines were used to treat 3000 patients per year.[10 11]

Data were also collected on any outpatient attendances or hospital admissions related to DMO or the treatments. Where an outpatient attendance was reported but no procedure undertaken, the average unit cost of an outpatient ophthalmology visit was used (varying between £80 and £101 per consultation depending on whether the consultation was 'non-consultant' vs 'consultant-led').[10 20] The CRFs recorded data on the grade of professional that attended the patient, for example, if a consultant attended to the patient, then the consultant-led unit cost was applied. Where a procedure was undertaken as part of the visit the relevant HRG code was derived using the HRG4+Reference Costs Grouper Software (NHS Digital, Leeds, UK).[10]

CRFs also recorded information on other tests or investigations, medication use including anti-VEGF/steroids or other rescue treatments. Anti-VEGF and steroid drugs were separately costed as these are considered an unbundled HRG. Costing of laser retreatments followed the same approach as costing for the index (first session) laser procedure.

Unit costs were derived from national compendia in accordance with NICE's Guide to the Methods of Technology Appraisal.[18] The key databases included the Department of Health and Social Care's Reference Costs 2018–2019 schedules,[20] the PSSRU's Unit Costs of Health and Social Care 2020 compendium,[19] 2020 volume of the British National Formulary.[21] Online supplemental table A1 gives a summary of the unit costs for resource use and the laser equipment. Resource inputs were valued by attaching unit costs.

The HRQoL of trial participants was assessed at baseline and at 12 and 24 months postrandomisation using the EQ-5D-5L instrument.[10] The EQ-5D-5L questionnaire was used to generate QALYs for the cost-effectiveness analysis.[22] The QALY is a measure that combines quantity and quality of life lived into a single metric, with one QALY equating to 1 year of full health. To convert EQ-5D-5L responses into health utility scores, we used the EQ-5D-5L Crosswalk Index Value Calculator which maps the EQ-5D-5L descriptive system data onto the EQ-5D-3L valuation set. This valuation set was recommended by NICE at the time when the analysis was undertaken.[22]

HRQoL was also assessed using two vision-specific measures: the National Eye Institute Visual Functioning Questionnaire-25 (NEI-VFQ-25) and Vision and Quality of Life Index (VisQoL).[23–25] The NEI-VFQ-25 is a validated questionnaire that has been used widely to evaluate visual outcomes in patients with eye diseases including diabetic retinopathy and DMO. The VisQol questionnaire has not been widely validated but is shorter than the NEI-VFQ-25 with only six attributes (physical well-being, independence, social well-being, self-actualisation, planning and organisation). The utilities for VisQoL were developed using a time-trade off exercise in people who were visually impaired which included patients with age-related macular degeneration, diabetic retinopathy and glaucoma.[25]

Summary statistics were generated for resource use variables, health utility values and QALYs by treatment allocation and assessment time point. For resource use and costs, mean values were compared between groups using two sample t-tests. Differences between groups, along with 95% CIs, were estimated using non-parametric bootstrap estimates (10 000 replications). For HRQoL, these were presented as mean values with their associated standard errors. Between-group differences were compared using the two-sample t-test.[10 11]

Multiple imputation by chained equations was used to predict missing health status (utility) scores and costs based on the assumption that data were missing at random.[10] Twenty imputed data sets were generated and used to inform the base-case analyses. Parameter estimates were pooled across the 20 imputed data sets using Rubin's rules to account for between-imputation and within-imputation components of variance terms associated with parameter estimates.[26]

The base-case cost-effectiveness analysis was performed using an intention-to-treat approach. Mean incremental costs and QALYs were estimated using seemingly unrelated regression (SUR) methods that account for the correlation between costs and outcomes. The SUR adjusted for covariates (baseline utilities, baseline body mass index, baseline BCVA, patient-reported previous use of anti-VEGF at baseline and previous use of macular laser).

## RESULTS

As reported elsewhere,[10 11] there was no difference in the clinical effectiveness of SML and SL. Table 1 shows that there were also no statistically significant differences between laser groups in EQ-5D-5L scores at baseline, 12 and 24 months.

Furthermore, there were no statistically significant differences between the two laser treatment groups for any of the VisQoL dimensions or NEI-VFQ-25 subscales at any follow-up time point (online supplemental tables A2 and A3, respectively).[10 11]

The mean numbers of laser treatments performed were 2.4 in the SML and 1.9 in the SL arm. This difference was statistically significant (p=0.002) (online supplemental table A4), but equated to less than one further session

**Table 1** EQ-5D-5L utility scores at baseline, 12 and 24 months and QALYs (base-case, imputed analysis)

| Variable | Subthreshold micropulse laser (n=133) Mean (SE) | Standard laser (n=132) Mean (SE) | Between-group difference (95% CI) | P value |
|---|---|---|---|---|
| EQ-5D Utility Scores* | | | | |
| Baseline | 0.758 (0.267) | 0.772 (0.226) | 0.014 (-0.074 to 0.046) | 0.640 |
| 12 months | 0.767 (0.250) | 0.758 (0.017) | 0.009 (-0.041 to 0.059) | 0.717 |
| 24 months | 0.739 (0.278) | 0.743 (0.279) | 0.004 (-0.064 to 0.056) | 0.897 |
| EQ-5D-5L QALYs | | | | |
| Over 2 years | 1.493 (0.024) | 1.485 (0.024) | 0.008 (-0.061 to 0.075) | 0.836 |

*Analysis adjusted for participant age, gender, baseline BCVA and participant's previous use of anti-VEGF and laser therapy at baseline, with repeated measures within participant and site.
BCVA, best-corrected visual acuity; QALYs, quality-adjusted life-years; VEGF, vascular endothelial growth factor.

of laser in the SML arm when compared with the SL arm during the 2-year trial. Furthermore, this difference was driven by a small number of participants requiring a higher number of lasers sessions in the SML group (13 participants required 6 or 7 laser treatments in the SML arm compared with 2 in the SL arm). Eighteen per cent of participants in the SML arm and 21% in the SL arm required rescue treatments in the study eye (almost all with anti-VEGF drugs; in addition, only one participant had a steroid injection). The average numbers of anti-VEGF treatments per arm from baseline to 24 months were 1.06 in the SML arm and 1.96 in the SL (online supplemental table A4). The number in the SL arm was skewed by five participants who received 10 or more injections.

Table 2 shows the NHS costs associated with resource use in the base-case (imputed) analysis by cost category and follow-up period. Costs of the first laser procedure are reported separately from those of subsequent laser retreatments. The total costs of laser therapy for each participant includes costs of the first laser procedure plus any subsequent laser retreatments they had. The mean (SE) cost for the first laser procedure (including costs of performing OCT and FFA to guide laser treatment, if done) was £45.59 (1.64) for the SML compared with £42.29 (1.69) for the SL; the difference was not statistically significant(p=0.09). The mean total NHS and PSS costs were lower in SML compared with the SL (£735.09

vs £1099.70) between baseline to 24 months postrandomisation; this difference was not statistically significant at the 5% level. The CIs around total costs were wide and overlapped and the difference was driven by the higher number of anti-VEGF rescue injections.

Over the 2-year follow-up period, participants in the SML arm, compared with the SL arm, experienced a non-statistically significant increase in QALYs of 0.008 (circa 3 days of good quality of life).[10 11] In addition, the mean NHS and PSS costs were lower in the SML arm compared with the SL arm (mean cost difference –£365) (see table 3). The CI for the cost difference was wide and ranged from cost saving to cost increasing. Although neither costs nor benefits were statistically significantly different between SML and SL, the incremental cost-effectiveness ratio (ICER) for the base-case imputed analysis indicated that SML dominates, as average costs for this intervention were slightly lower and average benefits marginally higher (but not significantly so) than those for SL. Assuming cost-effectiveness threshold of £20 000 per QALY, the probability that SML was cost-effective was 0.76, and the NMB associated with SML was positive. Figure 1 shows the joint distributions of costs and outcomes. The graph also highlights that SML has the potential to be cost saving as the majority of the bootstrapped iterations lie in the bottom half of the cost-effectiveness plane.

**Table 2** Economic costs by trial allocation arm and cost component category for the entire follow-up period in base-case (imputed) analysis (£, 2019–2020 prices)

| Parameter | Subthreshold micropulse laser Mean costs (SE) | Standard threshold laser Mean costs (SE) | Mean difference | Bootstrap 95% CI |
|---|---|---|---|---|
| Index laser procedure | 45.59 (1.64) | 42.29 (1.69) | 3.31 | (−1.33 to 7.95) |
| Laser retreatments | 53.02 (5.17) | 41.69 (4.42) | 11.32 | (−2.08 to 24.73) |
| Outpatient care | 124.85 (22.56) | 130.32 (31.58) | 5.47 | (−81.92 to 70.97) |
| Anti-VEGF drug costs | 511.63 (105.85) | 885.40 (183.29) | 373.77 | (−791.01 to 43.48) |
| Total NHS and PSS Costs | 735.09 (111.85) | 1099.70 (195.40) | 364.61 | (−807.09 to 77.87) |

NHS, National Health Service; PSS, personal social services; VEGF, vascular endothelial growth factor.

**Table 3** Cost-effectiveness, cost/QALY (£, 2020): SML compared with SL

| Mean incremental cost (95% CI) | Mean incremental QALY (95% CI) | ICER | Probability of cost-effectiveness* | Net monetary benefit* |
|---|---|---|---|---|
| Base-case analysis—ITT approach: Imputed attributable costs and QALYs, covariate adjusted† | | | | |
| 365 (−822 to 93) | 0.008 (−0.059 to 0.075) | Dominant | 0.763 | 520 (−925 to 1965) |

*Cost-effectiveness threshold is at £20 000/QALY threshold.
†Adjusted for baseline EQ-5D utility, BMI and minimisation variables at baseline (best corrected distance visual acuity and previous use of laser treatment).
BMI, body mass index; ICER, incremental cost-effectiveness ratio; ITT, intention-to-treat approach; QALY, quality-adjusted life-year; SL, standard laser; SML, subthreshold micropulse laser.

## DISCUSSION

DIAMONDS was a pragmatic clinical trial carried out in 16 NHS ophthalmology departments throughout the UK. The DIAMONDS trial was powered to detect not only differences in the primary outcome (BCVA) but also in important secondary outcomes (CRT and vision quality of life).[10 11] However, a limitation of this cost-effectiveness analysis is that we did not use BCVA as the primary outcome, instead we used QALYs as the main outcome. Participants were treated and followed as per routine clinical care. Costs and outcome data were collected prospectively. We found no significant differences in EQ-5D-5L and only a trivial non-significant difference of 0.008 in a calculation of QALYs. So, the verdict on whether one form of laser is better than the other shifts the focus onto the costs. This analysis found that costs were slightly higher (but not statistically significantly so) in the SL arm compared with the SML arm, due to more participants in the SL arm needing higher numbers (10 or more) of anti-VEGF rescue injections. Reporting both the costs and QALYs together in a ratio form, this meant the ICER for the base-case analysis indicated that SML is the dominant procedure, as average costs for this intervention were lower and average benefits were marginally higher than those for SL. However, caution should be taken when interpreting the results, given the wide CIs around the mean costs (which ranges from cost saving to cost increasing). Taking this into consideration, costs of SML and SL treatment arms seem comparable. We also conducted a per-protocol analysis as part of a sensitivity analysis as DIAMONDS was a non-inferiority trial. The results were in line with the intention-to-treat analysis, were SML remained the dominant option and the probability of of SML being cost-effectiveness at the £20k/QALY threshold was 0.773.

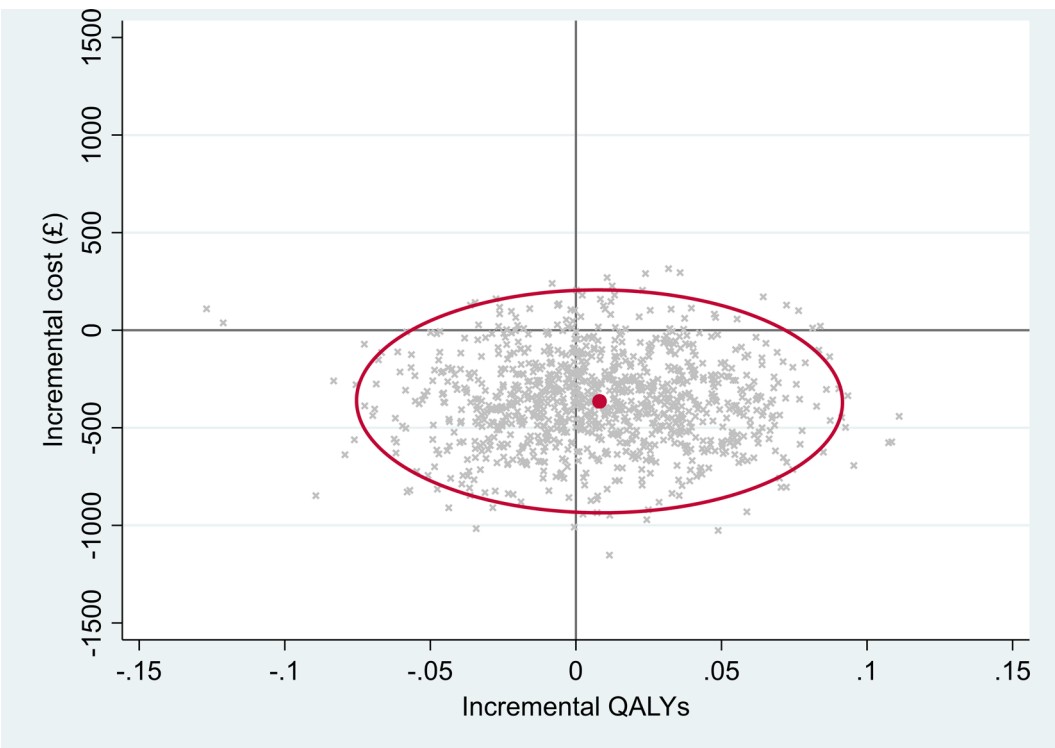

**Figure 1** Cost-effectiveness scatterplot with 95% confidence ellipses at 24 months for base-case within-trial analysis (NHS and PSS perspective, imputed, additionally controlled for baseline utilities). NHS, National Health Service; PSS, personal social services; QALYs, quality-adjusted life-years.

As noted in the main clinical effectiveness manuscript,[11] there may be advantages of SML over SL. Among these and importantly, the fact that SML does not cause any functional or structural damage to the retina[12–14] and, thus, can be repeated as needed. This is an advantage for patients as their retina will remain intact despite the application of SML but would lose cells as a result of the laser burn if SL is applied. Furthermore, the lack of a burn following SML makes the delivery of the treatment safer in less experienced hands as, unlike SL, it does not carry a risk of burning the fovea. Furthermore, for the same reason, it may allow training of allied non-medical staff to undertake this procedure, who currently, are already undertaking anti-VEGF injections in the UK and other parts of the world. This would help alleviate clinical capacity constraints experienced by most ophthalmology units across the world. The clinical effectiveness results reported previously[10 11] and the HRQoL results presented here showed that SML is as clinically effective as SL at a similar cost.

In DIAMONDS, two vision-specific patient-reported outcome quality-of-life instruments were used in addition to the generic preference-based EQ-5D-5L. Neither showed any statistically significant differences between laser arms. It has sometimes been mentioned that changes in vision that are sufficient to affect some activities of daily living, may not be reflected in changes in EQ-5D, because the EQ-5D may not be as sensitive to detect changes in quality of life as other visual-specific questionnaires such as the NEI-VFQ-25.[7 27 28] A mapping exercise from DIAMONDS data is underway to explore this important aspect.

One limitation in the current analysis is the fact that we had to use published costs of anti-VEGF drugs (at list prices, these can range from £5 to £7k a year),[7–9] rather than costs to the NHS. We know that there are confidential price discounts for these drugs when used in the NHS (without taking into account the administration of the injection into the eye and monitoring visits) and this will most likely reduce the cost differential between the two arms.[7–9] Another limitation is that DIAMONDS included only patients with CRT<400μ, so extrapolations with regard to the cost-effectiveness of SML when compared with SL in thicker retinas, where SL is known to be less effective, cannot be made. Furthermore, the analysis was conducted from an NHS and PSS perspective, so we have not taken into account any broader societal costs such as time off work or care for children or other dependents, when patients have laser treatment.

Both SML and SL were successful in 80% of participants.[11] Macular laser treatment is known to be less expensive than anti-VEGF drugs and may be more convenient and acceptable to participants, although no trials have been conducted comparing the cost-effectiveness of macular laser with anti-VEGFs in people with <400μ CRT DMO. Thus, SML should be considered as first line therapy for patients with central involving DMO with CRT of <400μ.

**Author affiliations**
$^1$Warwick Clinical Trials Unit, Warwick Medical School, University of Warwick, Coventry, UK
$^2$University Hospitals Coventry and Warwickshire NHS Trust, Coventry, UK
$^3$Northern Ireland Clinical Trials Unit, Belfast, UK
$^4$Department of Ophthalmology, Wellcome-Wolfson Institute for Experimental Medicine, Queens University Belfast, Belfast, UK

**Acknowledgements** The authors would like to thank all the DIAMONDS participants and the DIAMONDS patient and public involvement (PPI) group. We also thank Dr Pamela Royle for literature searches. We would also like to thank Iridex for providing a free loan for the laser equipment for the trial and William Moore, Joan Stauffer, George Marcellino (in memoriam), Gareth Hymas, Nick Fitrzyk and Suzanne Kelly, for their support. Iridex had no role in the design, data analysis or interpretation of the DIAMONDS data or in the writing of this report. We thank also the NIHR HTA programme for providing funding for the trial and note that the views expressed in this paper are those of the authors and not necessarily those of the NHS, the NIHR or the Department of Health and Social Care.

**Collaborators** Noemi Lois, Queen's University and Royal Victoria Hospital, Belfast H&SC Trust; Augusto Azuara-Blanco, Queen's University and Royal Victoria Hospital, Belfast H&SC Trust; Christina Campbell, NICTU, Belfast; Danny McAuley, Queen's University and Royal Victoria Hospital, Belfast H&SC Trust; Mandy Maredza, Warwick University; Hema Mistry, Warwick University; Norman Waugh, Warwick University; Nachiketa Acharya, Sheffield Teaching Hospitals NHS Foundation Trust; Tariq M Aslam, Manchester Royal Eye Hospital, Central Manchester University Hospitals NHS Foundation Trust; Clare Bailey, Bristol Eye Hospital, University Hospitals Bristol NHS Foundation Trust; Victor Chong, Royal Free Hospital NHS Foundation Trust, London; Louise Downey, Hull and East Yorkshire Hospital, Hull and East Yorkshire NHS Trust; Haralabos Eleftheriadis, Kings College Hospital NHS Foundation Trust; Samia Fatum, John Radcliffe Hospital, Oxford University Hospitals NHS Foundation Trust; Sheena George, Hillingdon Hospitals NHS Foundation Trust; Faruque Ghanchi, Bradford Teaching Hospitals NHS Trust; Markus Groppe, Stoke Mandeville Hospital, Buckinghamshire NHS Trust; Robin Hamilton, Moorfields Eye Hospital NHS Foundation Trust; Geeta Menon, Frimley Park Hospital NHS Foundation Trust; Ahmed Saad, James Cook University Hospital, South Tees Hospitals NHS Foundation Trust; Sobha Sivaprasad, Moorfields Eye Hospital NHS Foundation Trust; Marianne Shiew, Hinchingbrooke Hospital North West Anglia NHS Trust; David H Steel, Sunderland Eye Infirmary, City Hospitals Sunderland NHS Foundation Trust; James Stephen Talks, Newcastle Eye Centre, Royal Victoria Infirmary, Newcastle upon Tyne Hospitals NHS Foundation Trust; Catherine Adams, NICTU, Belfast; Paul Doherty, NICTU, Belfast; Evie Gardener, NICTU, Belfast; Aby Joseph, NICTU, Belfast; Cliona McDowell, NICTU, Belfast; Matthew Mills, NICTU, Belfast; Mike Clarke, Queens University, Belfast.

**Contributors** HM contributed to the design of the health economics plan and provided oversight of all aspects of the economic evaluation including its design, conduct, analysis and reporting of this paper. MM conducted the analysis and reporting of the economic evaluation and contributed to the drafting of this paper. CC performed the statistical analyses, assistedwith interpretation of data and contributed to the drafting of this paper. NL is the Chief Investigator for the DIAMONDS trial, she is overall guarantor, she conceived the trial, with input from the DIAMONDS Study Group, and led it to its successful completion. In addition, contributed with the identification, recruitment, examination and treatment of all participants enrolled at the Belfast site, as well as to data collection, analysis and interpretation of the trial findings. She contributed to the drafting of this paper.

**Funding** DIAMONDS was funded by the Health Technology Assessment (HTA) programme of the National Institute for Health Research (NIHR), United Kingdom NIHR-HTA (13/142/04).

**Disclaimer** The NIHR had no role in the design or conduct of this research. Iridex provided a free loan for the laser equipment for the trial. Iridex had no role in the design or conduct of the trial or on data analysis or interpretation of the DIAMONDS data or writing of this report.

**Competing interests** None declared.

**Patient and public involvement** Patients and/or the public were involved in the design, or conduct, or reporting, or dissemination plans of this research. Refer to the Methods section for further details.

**Patient consent for publication** Not applicable.

**Ethics approval** The protocol for the DIAMONDS trial was approved by the Office for Research Ethics Committees Northern Ireland (ORECNI 15/NI/0197). A Clinical

Trial Authorisation (CTA) was obtained from the Medicines and Healthcare products Regulatory Agency (MHRA) (32485/0029/001-0001).

**Provenance and peer review** Not commissioned; externally peer reviewed.

**Data availability statement** Data are available on reasonable request. The datasets used during the current study are available from the corresponding author upon reasonable request.

**ORCID iD**
Hema Mistry http://orcid.org/0000-0002-5023-1160

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
