## [Reviewer comments · BMJ Open]

ARTICLE DETAILS

TITLE (PROVISIONAL)	Subthreshold micropulse laser versus standard laser for the treatment of central-involving diabetic macular oedema with central retinal thickness of <400µ: a cost-effectiveness analysis from the DIAMONDS trial
AUTHORS	Mistry, Hema; Maredza, Mandy; Campbell, Christina; Lois, Noemi; Study Group, DIAMONDS

VERSION 1 – REVIEW

REVIEWER	Iglicki, Matius University of Buenos Aires
REVIEW RETURNED	20-Sep-2022

GENERAL COMMENTS	Mistry, Hema et al. present an interesting study that evaluates Subthreshold micropulse laser versus standard laser for the treatment of central-involving diabetic macular oedema with central retinal thickness of <400µ: a cost-effectiveness analysis from the DIAMONDS trial This Manuscript suggest that there were no statistically significant differences in EQ-5D-5L scores or costs between SML and SL. Given these findings and the fact that SML does not burn the retina, unlike SL, and has equivalent efficacy to SL, it may be preferred for the treatment of people with DMO with CRT <400µ. Besides how the magnitude of these data add new findings compare to the current standard can not be determined based on this study The results are encouraging and further study is warranted. here some relevant points : 1. please add on the keywords this does not match with the manuscript2. The authors should express why is relevant for DME patients to evaluate Subthreshold micropulse laser versus standard laser for the treatment of central-involving diabetic macular oedema with central retinal thickness of <400µ: a cost-effectiveness What does it Change for the current standard of care ? 3. The authors should explain why their findings make a different for ophthalmologist around the world and for the readers of BMJ Open
------------------	--

4. The authors should explain the source of the information and what were the criteria they used for adding to the paper Were the assessors masked? What was the ICC between them in order to analyzed the data ? I Was the randomization digitalized? . If not please added

5. Please add references and rephrase the sentence. English grammar should be applied.

“As noted in the main clinical effectiveness manuscript there may be advantages of SML over SL, including the fact that SML does not cause any structural changes in the retina and, thus, can be repeated potentially indefinitely, as needed. This makes it also potentially safer in less experienced hands. Furthermore, for the same reason, it may allow training of non-specialist/non-medical staff who, currently, are already undertaking anti-VEGF injections in the UK and other parts of the world. This would help alleviating NHS capacity constraints. The clinical effectiveness results reported previously (7) and the HRQoL results presented here showed that SML is as clinically effective as SL at a similar cost.”

6. please add in the introduction papers which have been published showing the important for DME and RD patients to performed the test ancillary test such us OCTs , wield Field Retinography and add how this will help physician around the world to proper diagnosis not only for DME but also for macular deseases , add one line in the introduction of this and also in the discussion section These papers should be describe in this general considerations and in the reference

Reference these :

- EYE 2022 Jan 18. doi: 10.1038/s41433-022-01931-9. Online ahead of print.

Vitrectomized vs non-vitrectomized eyes in DEX implant treatment for DMO-Is there any difference? the VITDEX study

-

Multicenter Study

Br J Ophthalmol.2020 May;104(5):666-671. doi:

10.1136/bjophthalmol-2019-314523. Epub 2019 Aug 7.

Outer retinal hyperreflective deposits (ORYD): a new OCT feature in naïve diabetic macular oedema after PPV with ILM peeling

-Retina. 2019 Jan;39(1):44-51. doi: 10.1097/IAE.0000000000002196.
DEXAMETHASONE IMPLANT FOR DIABETIC MACULAR EDEMA IN NAIVE COMPARED WITH REFRACTORY EYES: The International Retina Group Real-Life 24-Month Multicenter Study. The IRGREL-DEX Study.
Iglicki M1, Busch C2, Zur D3,4, Okada M5, Mariussi M6, Chhablani JK7, Cebeci Z8, Fraser-Bell S9, Chaikitmongkol V10, Couturier A11, Giacipoli E12, Lupidi M13, Rodríguez-Valdés PJ14, Rehak M2, Fung AT15,16,17, Goldstein M3,4, Loewenstein A3,4,17.

-Acta Diabetol. 2018 Jun;55(6):541-547. doi: 10.1007/s00592-018-1117-z. Epub 2018 Mar 1.
Progression of diabetic retinopathy severity after treatment with dexamethasone implant: a 24-month cohort study the 'DR-Pro-DEX Study'.
Iglicki M1, Zur D2,3, Busch C4, Okada M5, Loewenstein A2,3,6.

-Ophthalmologica. 2019;241(1):9-16. doi: 10.1159/000492132. Epub 2018 Nov 8.
Effectiveness and Safety of Intravitreal Dexamethasone Implant (Ozurdex) in Patients with Diabetic Macular Edema: A Real-World Experience.
Mello Filho P1, Andrade G2, Maia A1, Maia M1, Biccás Neto L1, Muralha Neto A3, Moura Brasil O4, Minelli E5, Dalloul C6, Iglicki M7.

-Ophthalmic Res. 2019 May 2:1-6. doi: 10.1159/000499540. [Epub ahead of print]
The Role of Steroids in the Management of Diabetic Macular Edema.
Zur D1, Iglicki M2, Loewenstein A3.

-Acta Diabetol. 2019 Oct;56(10):1141-1147. doi: 10.1007/s00592-019-01357-y. Epub 2019 May 14.
TRActional DIabetic reTInal detachment surgery with co-adjuvant intravitreal dexamethasONE implant: the TRADITION STUDY.

-Iglicki M1, Zur D2,3, Fung A4,5,6, Gabrielle PH7, Lupidi M8, Santos R9, Busch C10, Rehak M10, Cebeci Z11, Charles M12, Masarwa D2,3, Schwarz S2,3, Barak A2,3, Loewenstein A2,3,13; International Retina Group (IRG).

-Acta Ophthalmol. 2019 Aug 17. doi: 10.1111/aos.14230. [Epub ahead of print]

Disorganization of retinal inner layers as a biomarker in patients with diabetic macular oedema treated with dexamethasone implant.

Zur D1, Iglicki M2, Sala-Puigdollers A3, Chhablani J4, Lupidi M5, Fraser-Bell S6, Mendes TS7,8, Chaikitmongkol V9, Cebeci Z10, Dollberg D1, Busch C11, Invernizzi A12,13, Habot-Wilner Z1, Loewenstein A1,14; International Retina Group (IRG).

-PLoS One. 2018; 13(7): e0200365.

Published online 2018 Jul 11. doi:

10.1371/journal.pone.0200365PMCID: PMC6040739PMID:

29995929 Biomarkers and predictors for functional and anatomic outcomes for small gauge pars plana vitrectomy and peeling of the internal limiting membrane in naïve diabetic macular edema: The VITAL Study

-Acta Diabetol

2020 Apr 16. doi: 10.1007/s00592-020-01530-8. Online ahead of print.

Results in comparison between 30 gauge ultrathin wall and 27 gauge needle in sutureless intraocular lens flanged technique in diabetic patients: 24-month follow-up study

Matias Iglicki 1 , Dinah Zur 2 , Hermino Pablo Negri 3 , Joaquin Esteves 3 , Romina Arias 3 , Emanuel Holsman 3 , Anat Loewenstein 2 , Catharina Busch 4

-Acta Ophthalmol

. 2020 Mar;98(2):e217-e223. doi: 10.1111/aos.14230. Epub 2019 Aug 17.

Disorganization of retinal inner layers as a biomarker in patients with diabetic macular oedema treated with dexamethasone implant
Dinah Zur # 1 , Matias Iglicki # 2 , Anna Sala-Puigdollers 3 , Jay Chhablani 4 , Marco Lupidi 5 , Samantha Fraser-Bell 6 , Thais Sousa Mendes 7 8 , Voraporn Chaikitmongkol 9 , Zafer Cebeci 10 , Dolev Dollberg 1 , Catharina Busch 11 , Alessandro Invernizzi 12 13 , Zohar Habot-Wilner 1 , Anat Loewenstein 1 14 , International Retina Group (IRG)

-

Multicenter Study

-Acta Diabetol . 2019 Oct;56(10):1141-1147. doi: 10.1007/s00592-019-01357-y. Epub 2019 May 14.

TRActional DIabetic reTInal detachment surgery with co-adjunct intravitreal dexamethasONE implant: the TRADITION STUDY

Matias Iglicki 1 , Dinah Zur 2 3 , Adrian Fung 4 5 6 , Pierre-Henry Gabrielle 7 , Marco Lupidi 8 , Rodrigo Santos 9 , Catharina Busch 10 , Matus Rehak 10 , Zafer Cebeci 11 , Martin Charles 12 , Dua Masarwa 2 3 , Shulamit Schwarz 2 3 , Adiel Barak 2 3 , Anat Loewenstein 2 3 13 , International Retina Group (IRG)

Affiliations expand

PMID: 31089929 DOI: 10.1007/s00592-019-01357-y

-Observational Study

Ophthalmologica

. 2019;241(1):9-16. doi: 10.1159/000492132. Epub 2018 Nov 8.
Effectiveness and Safety of Intravitreal Dexamethasone Implant
(Ozurdex) in Patients with Diabetic Macular Edema: A Real-World
Experience

Paulo Mello Filho 1 , Gabriel Andrade 2 , Andre Maia 1 , Mauricio
Maia 1 , Laurentino Biccás Neto 1 , Acacio Muralha Neto 3 ,
Oswaldo Moura Brasil 4 , Eduardo Minelli 5 , Claudio Dalloul 6 ,
Matias Iglicki

Eye 2021 Aug 9. doi: 10.1038/s41433-021-01722-8. Online ahead
of print.

Next-generation anti-VEGF agents for diabetic macular oedema
Matias Iglicki # 1 , David Pérez González # 2 , Anat Loewenstein 2
, Dinah Zur 2

Affiliations expand

PMID: 34373607 DOI: 10.1038/s41433-021-01722-8

-Retina. 2019 Nov;39(11):2161-2166. doi:
10.1097/IAE.0000000000002270.

UNDERDIAGNOSED OPTIC DISK PIT MACULOPATHY: Spectral
Domain Optical Coherence Tomography Features For Accurate
Diagnosis

Matias Iglicki 1 , Catharina Busch 2 , Anat Loewenstein 3 4 5 ,
Adrian T Fung 6 7 8 , Alessandro Invernizzi 8 9 , Miriana Mariussi
10 , Romina Arias 1 , Pierre-Henry Gabrielle 11 , Zafer Cebeci 12 ,
Mali Okada 13 , Jerzy Nawrocki 14 15 , Zofia Michalewska 14 15 ,
Michaella Goldstein 3 4 , Adiel Barak 3 4 , Dinah Zur 3

-EYE-2021 Apr;35(4):1111-1116. doi: 10.1038/s41433-020-01309-
9. Epub 2021 Jan 11.

Longer-acting treatments for neovascular age-related macular
degeneration-present and future

-Ophthalmology.2018 Feb;125(2):267-275. doi:

10.1016/j.ophtha.2017.08.031. Epub 2017 Sep 19.

OCT Biomarkers as Functional Outcome Predictors in Diabetic
Macular Edema Treated with Dexamethasone Implant

- Br J Ophthalmol 2020 Dec 7;bjophthalmol-2020-317422. doi:

10.1136/bjophthalmol-2020-317422. Online ahead of print.

Central serous chorioretinopathy imaging biomarkers

-Eye .2022 Feb;36(2):273-277. doi: 10.1038/s41433-021-01722-8.
Epub 2021 Aug 9.

Next-generation anti-VEGF agents for diabetic macular oedema

-Retina2022 Mar 2. doi: 10.1097/IAE.0000000000003466. Online
ahead of print.'Target Sign' - A near infrared feature and
multimodal imaging in a pluri-ethnic cohort with RDH5-related
fundus albipunctatus7.please add how, and how long takes for a

	retina specialist to think about OCT biomarker to star the proper treatment -Ophthalmol Retina 2021 Nov;5(11):1097-1106. doi: 10.1016/j.oret.2021.01.013. Epub 2021 Feb 1.Detection of Diabetic Retinopathy from Ultra-Widefield Scanning Laser Ophthalmoscope Images: A Multicenter Deep Learning Analysis8. Results could be misinterpreted add a short summary of the similarities in different devices where can be added the algorithms and also add diferente oct modalities -Eye .2022 Aug 29. doi: 10.1038/s41433-022-02222-z. Online ahead of print. Naïve subretinal haemorrhage due to neovascular age-related macular degeneration. pneumatic displacement, subretinal air, and tissue plasminogen activator: subretinal vs intravitreal aflibercept- the native study Please apply correction for misspelling and English grammar . This paper should be corrected by an english redactor
--	--

REVIEWER	Valera-Cornejo, Diego Alejandro Instituto Mexicano De Oftalmologia, Retina
REVIEW RETURNED	06-Oct-2022

GENERAL COMMENTS	It is a well-written manuscript that analyzes the cost-effectiveness between SL and SML. It would be appropriate to discuss and compare other studies that evaluate costs and utilities regarding laser treatment and SML for DME. Page 15 line 14 ... potentially indefinitely, as needed... I would recommend avoiding stating that the treatment would be indefinite.
------------------	---

REVIEWER	Quarta , Alberto University G. D'Annunzio Chieti-Pescara, Ophthalmology
REVIEW RETURNED	01-Feb-2023

GENERAL COMMENTS	The work seems complete and exhaustive, well written for methods, results and discussion, with great data presentation and analysis. The study could be eligible for publication.
------------------	---

REVIEWER	Ho, Mary The Chinese University of Hong Kong
REVIEW RETURNED	13-Jun-2023

GENERAL COMMENTS	The study topic of analyzing the costs difference between SML (subthreshold micropulse laser) and SL (standard laser) treatments for Diabetic Macular Edema is interesting and relevant, as both treatments are commonly used in clinical practice. The study builds upon the results of a well-conducted clinical trial (DIAMONDS) that compared the efficacy of SML, and SL treatments for DME. The study's focus on cost-effectiveness is especially important given the increasing pressure on healthcare systems to deliver high-quality care while containing costs. Before proceeding further, a few points need to be clarified to ensure that the study's results and conclusions are reliable and valid:
------------------	--

	Comment 1: The results of the analysis presented in the report are somewhat confusing and difficult to interpret. Furthermore, the confidence interval for the estimated cost-effectiveness ratios is relatively wide, which indicates a high degree of uncertainty around the results. However, the report suggests that the costs of the SML (subthreshold micropulse laser) and SL (standard laser) treatments are comparable. Similarly, the tables presented in the report suggest that SML may be more cost-effective than SL, despite the similar QALY (quality-adjusted life year) estimates. This finding may be counterintuitive and warrants further investigation to understand the potential reasons for this unexpected result. Comment 2: Table 2 in the report presents the cost estimates for the SML (subthreshold micropulse laser) and SL (standard laser) treatments, and it is important to note that the cost difference between the two treatments depends on several factors. These include whether the consultation is led by a consultant, the number of rescue treatments required, and the number of anti-VEGF injections administered. However, the results for the number of anti-VEGF injections should be viewed with caution, as they are heavily skewed by a small number of patients in the SL group who received more than 10 injections. The factor on consultant led / non-consultant led clinic is probably a random factor, and should be excluded in the analysis. Furthermore, the published costs for anti-VEGF agents were used, which may not accurately reflect the actual costs incurred by the NHS. As a result, the reported cost difference between the SML and SL treatments may be exaggerated.
--	---

REVIEWER	Müller, Dirk University Hospital Cologne, Institute of Health Economics and Clinical Epidemiology
REVIEW RETURNED	01-Aug-2023

GENERAL COMMENTS	Overall, the analysis by Mistry et al. which performed a cost-effectiveness analysis from the DIAMONDS trial fulfills most of the core recommendations for conducting economic analyses alongside a clinical trial. However, before publication can be recommended, some aspects have to be clarified and improved.  • Page 5, line 5-8: the authors state that their economic study is based on data obtained from an appropriately powered pragmatic randomised clinical trial. This might be true for the clinical endpoint (BCVA) but is not for the issue of cost-effectiveness. Whereas it is common to base a trial-based cost-effectiveness study on the sample size of the trial, ideally, researchers should set up a formal hypothesis for the cost-effectiveness analysis (to avoid underpowerment, see Ramsey et al. Value in Health 2015). This should be stated as a limitation, particularly because the primary study endpoint was not used here. • Page 9, line 25-50: the authors state that the EQ-5D was used for generating QALYs. In the same paragraph the NEI-VFQ-25 and the VisQoL are mentioned, with the latter being used for a time-trade off exercise. Please provide more information about the necessity to add these two instruments for assessing HRQoL. QALYs were derived from the EQ-5D and I would have expected
------------------	--

	information about the tariffs used for valuing these QALYs obtained from the study. I do not understand how the NEI-VFQ-25 and the VisQoL were incorporated in the analysis. Did the tables A2 and A3 play any role for the analysis?  • Bootstrap: a graphical illustration would be useful. • Discussion: overall, this section is disappointing. Several aspects potentially affecting the internal and external validity of the analysis should be discussed more extensively (e.g., performing an ITT-analysis for the economic evaluation but a PP-analysis for the trial, or the impact of calculating a protocol-driven resource use). • Supplemental material: the authors added the CHEERS-checklist for demonstrating reporting quality which should be mentioned in the method section (with reference). In addition, it would be useful to consider the work of Ramsey et al. for assuring the methodological quality (Ramsey et al. 2005 and 2015). • Supplemental material (CHEERS-checklist): many of the items are responded with N/A. At least for item 20 and 24 (uncertainty) this should be updated because uncertainty was analyzed (e.g., bootstrap). For others (e.g., item 18) one could argue that the items were not addressed (but could have). • It is conspicuous that only a small number of references was used for the study and no additional references in the discussion to put study results in a broader context. Please revise.
--	--

VERSION 1 – AUTHOR RESPONSE

Reviewers comments	Our response
Reviewer 1 Mistry, Hema et al. present an interesting study that evaluates Subthreshold micropulse laser versus standard laser for the treatment of central-involving diabetic macular oedema with central retinal thickness of <400µ: a cost-effectiveness analysis from the DIAMONDS trial This Manuscript suggest that there were no statistically significant differences in EQ-5D-5L scores or costs between SML and SL. Given these findings and the fact that SML does not burn the retina, unlike SL, and has equivalent efficacy to SL, it may be preferred for the treatment of people with DMO with CRT <400µ. Besides how the magnitude of these data add new findings compare to the current standard can not be determined based on this study The results are encouraging and further study is warranted. here some relevant points :	Thank you for your comment. No response needed.
1. please add on the keywords this does not match with the manuscript	We have updated the keywords on the manuscript to match the ones on the online system.
2. The authors should express why is relevant for DME patients to evaluate Subthreshold micropulse laser versus standard laser for the treatment of central-involving diabetic macular oedema with central retinal thickness of <400µ: a cost-effectiveness	We thank the Reviewer for this suggestion. In order to address this issue, we have added to the Introduction section of the manuscript (second paragraph): “DIAMONDS found SML and SL to have equivalent clinical efficacy.(7, 8) This finding is clinically important given the fact

	that SML, unlike SL, does not cause any functional or structural damage to the retina (9-11) and, thus, may be preferred by patients and doctors. Herein, we present... Please see also our response below with additional information we have added on this regard to the Discussion section of the manuscript.
3. The authors should explain why their findings make a difference for ophthalmologists around the world and for the readers of BMJ Open	We thank the reviewer for this suggestion. However, we think we have covered this already in the Discussion section of the manuscript, where we stated (Discussion section, second paragraph): “...there may be advantages of SML over SL, including the fact that SML does not cause any structural changes in the retina and, thus, can be repeated as needed. This makes it also safer in less experienced hands. Furthermore, for the same reason, it may allow training of non-specialist/non-medical staff who, currently, are already undertaking anti-VEGF injections in the UK and other parts of the world. This would help alleviate NHS capacity constraints. The clinical effectiveness results reported previously (7, 8) and the HRQoL results presented here showed that SML is as clinically effective as SL at a similar cost.” To make the above text more generalisable to other countries, however, we have removed now the specific reference to the NHS, as follows: “Furthermore, for the same reason, it may allow training of non-specialist/non-medical staff who, currently, are already undertaking anti-VEGF injections in the UK and other parts of the world. This would help alleviate clinical capacity constraints experienced by most ophthalmology units across the world.”
4. The authors should explain the source of the information and what were the criteria they used for adding to the paper. Were the assessors masked? What was the ICC between them in order to analyze the data? Was the randomization digitalized? If not please add	We thank the reviewer for this comment and suggestion. We think, however, that the purpose of the current paper is to provide detail and specific information about the cost-effectiveness evaluation undertaken in DIAMONDS. All methodological aspects raised by the Reviewer, although of great importance, have been all addressed in the main clinical effectiveness paper already published (Lois et al, Ophthalmology. 2023 Jan;130(1):14-2).
5. Please add references and rephrase the sentence. English grammar should be applied. “As noted in the main clinical effectiveness manuscript there may be advantages of SML over SL, including the fact that SML does not cause any structural changes in the retina and, thus, can be repeated potentially indefinitely, as needed. This makes it also potentially safer in less experienced hands. Furthermore, for the	We thank the Reviewer for this suggestion. We have now rephrased the sentence and added pertinent references, as follows: “As noted in the main clinical effectiveness manuscript (7) there may be advantages of SML over SL. Among these and importantly, the fact that SML does not cause any functional or structural damage to the retina and, thus, can be repeated as needed. This is an advantage

same reason, it may allow training of non-specialist/non-medical staff who, currently, are already undertaking anti-VEGF injections in the UK and other parts of the world. This would help alleviating NHS capacity constraints. The clinical effectiveness results reported previously (7) and the HRQoL results presented here showed that SML is as clinically effective as SL at a similar cost.“	for patients as their retina will remain intact despite the application of SML but would lose cells as a result of the laser burn if SL is applied. Furthermore, the lack of a burn following SML makes the delivery of the treatment safer in less experienced hands as, unlike SL, it does not carry a risk of burning the fovea. Furthermore, for the same reason, it may allow training of allied non-medical staff to undertake this procedure, who currently are already undertaking anti-VEGF injections in the UK and other parts of the world. This would help alleviate the capacity constraints experienced by most ophthalmology units across the world. The clinical effectiveness results reported previously (7, 8) and the HRQoL results presented here showed that SML is as clinically effective as SL at a similar cost.”
6. please add in the introduction papers which have been published showing the important for DME and RD patients to performed the test ancillary test such us OCTs, wield Field Retinography and add how this will help physician around the world to proper diagnosis not only for DME but also for macular diseases, add one line in the introduction of this and also in the discussion section These papers should be describe in this general considerations and in the reference Reference these : - EYE 2022 Jan 18. doi: 10.1038/s41433-022-01931-9. Online ahead of print. Vitrectomized vs non-vitrectomized eyes in DEX implant treatment for DMO-Is there any difference? the VITDEX study - Multicenter Study Br J Ophthalmol.2020 May;104(5):666-671. doi: 10.1136/bjophthalmol-2019-314523. Epub 2019 Aug 7. Outer retinal hyperreflective deposits (ORYD): a new OCT feature in naïve diabetic macular oedema after PPV with ILM peeling -Retina. 2019 Jan;39(1):44-51. doi: 10.1097/IAE.0000000000002196. DEXAMETHASONE IMPLANT FOR DIABETIC MACULAR EDEMA IN NAIVE COMPARED WITH REFRACTORY EYES: The International Retina Group Real-Life 24-Month Multicenter Study. The IRGREL-DEX Study. Igllicki M1, Busch C2, Zur D3,4, Okada M5, Mariussi M6, Chhablani JK7, Cebeci Z8, Fraser-Bell S9, Chaikitmongkol V10, Couturier A11, Giancipoli E12, Lupidi M13, Rodríguez-Valdés PJ14, Rehak M2, Fung AT15,16,17, Goldstein M3,4, Loewenstein A3,4,17. -Acta Diabetol. 2018 Jun;55(6):541-547. doi: 10.1007/s00592-018-1117-z. Epub 2018 Mar 1.	We thank the Reviewer once again for this comment. We do not think, however, that an in depth discussion about how to diagnose macular oedema would be pertinent for this manuscript presenting a cost-utility analysis of a trial on DMO. The list of references suggested is exhaustive but, equally, we do not think pertinent. Instead we have added in some more relevant references with regards to SML and SL in the Introduction and Discussion section. We have checked the manuscript throughout for English spelling and grammar and corrected any errors.

Progression of diabetic retinopathy severity after treatment with dexamethasone implant: a 24-month cohort study the 'DR-Pro-DEX Study'. Iglicki M1, Zur D2,3, Busch C4, Okada M5, Loewenstein A2,3,6.

-Ophthalmologica. 2019;241(1):9-16. doi: 10.1159/000492132. Epub 2018 Nov 8. Effectiveness and Safety of Intravitreal Dexamethasone Implant (Ozurdex) in Patients with Diabetic Macular Edema: A Real-World Experience. Mello Filho P1, Andrade G2, Maia A1, Maia M1, Biccias Neto L1, Muralha Neto A3, Moura Brasil O4, Minelli E5, Dalloul C6, Iglicki M7.

-Ophthalmic Res. 2019 May 2:1-6. doi: 10.1159/000499540. [Epub ahead of print] The Role of Steroids in the Management of Diabetic Macular Edema. Zur D1, Iglicki M2, Loewenstein A3.

-Acta Diabetol. 2019 Oct;56(10):1141-1147. doi: 10.1007/s00592-019-01357-y. Epub 2019 May 14. TRActional Diabetic reTInal detachment surgery with co-adjutant intravitreal dexamethasONE implant: the TRADITION STUDY. -Iglicki M1, Zur D2,3, Fung A4,5,6, Gabrielle PH7, Lupidi M8, Santos R9, Busch C10, Rehak M10, Cebeci Z11, Charles M12, Masarwa D2,3, Schwarz S2,3, Barak A2,3, Loewenstein A2,3,13; International Retina Group (IRG).

-Acta Ophthalmol. 2019 Aug 17. doi: 10.1111/aos.14230. [Epub ahead of print] Disorganization of retinal inner layers as a biomarker in patients with diabetic macular oedema treated with dexamethasone implant. Zur D1, Iglicki M2, Sala-Puigdollers A3, Chhablani J4, Lupidi M5, Fraser-Bell S6, Mendes TS7,8, Chaikitmongkol V9, Cebeci Z10, Dollberg D1, Busch C11, Invernizzi A12,13, Habot-Wilner Z1, Loewenstein A1,14; International Retina Group (IRG).

-PLoS One. 2018; 13(7): e0200365. Published online 2018 Jul 11. doi: 10.1371/journal.pone.0200365PMCID: PMC6040739PMID: 29995929 Biomarkers and predictors for functional and anatomic outcomes for small gauge pars plana vitrectomy and peeling of the internal limiting membrane in naïve diabetic macular edema: The VITAL Study

-Acta Diabetol 2020 Apr 16. doi: 10.1007/s00592-020-01530-8. Online ahead of print. Results in comparison between 30 gauge ultrathin wall and 27 gauge needle in sutureless

intraocular lens flanged technique in diabetic patients: 24-month follow-up study. Matias Iglicki 1, Dinah Zur 2, Hermino Pablo Negri 3, Joaquin Esteves 3, Romina Arias 3, Emanuel Holsman 3, Anat Loewenstein 2, Catharina Busch 4

-Acta Ophthalmol 2020 Mar;98(2):e217-e223. doi: 10.1111/aos.14230. Epub 2019 Aug 17. Disorganization of retinal inner layers as a biomarker in patients with diabetic macular oedema treated with dexamethasone implant Dinah Zur # 1, Matias Iglicki #2, Anna Sala-Puigdollers 3, Jay Chhablani 4, Marco Lupidi 5, Samantha Fraser-Bell 6, Thais Sousa Mendes 7 8, Voraporn Chaikitmongkol 9, Zafer Cebeci 10, Dolev Dollberg 1, Catharina Busch 11, Alessandro Invernizzi 12 13, Zohar Habot-Wilner 1, Anat Loewenstein 1 14, International Retina Group (IRG)

Multicenter Study -Acta Diabetol . 2019 Oct;56(10):1141-1147. doi: 10.1007/s00592-019-01357-y. Epub 2019 May 14. TRActional Diabetic reTInal detachment surgery with co-adjuvant intravitreal dexamethasONE implant: the TRADITION STUDY Matias Iglicki 1, Dinah Zur 2 3, Adrian Fung 4 5 6, Pierre-Henry Gabrielle 7, Marco Lupidi 8, Rodrigo Santos 9, Catharina Busch 10, Matus Rehak 10, Zafer Cebeci 11, Martin Charles 1, Dua Masarwa 2 3, Shulamit Schwarz 2 3, Adiel Barak 2 3, Anat Loewenstein 2 3 13, International Retina Group (IRG) Affiliations expand PMID: 31089929 DOI: 10.1007/s00592-019-01357-y

-Observational Study Ophthalmologica 2019;241(1):9-16. doi: 10.1159/000492132. Epub 2018 Nov 8. Effectiveness and Safety of Intravitreal Dexamethasone Implant (Ozurdex) in Patients with Diabetic Macular Edema: A Real-World Experience Paulo Mello Filho 1, Gabriel Andrade 2, Andre Maia 1, Mauricio Maia 1, Laurentino Biccias Neto 1, Acacio Muralha Neto 3, Oswaldo Moura Brasil 4, Eduardo Minelli 5, Claudio Dalloul 6, Matias Iglicki

Eye 2021 Aug 9. doi:10.1038/s41433-021-01722-8. Online ahead of print. Next-generation anti-VEGF agents for diabetic macular oedema Matias Iglicki #1, David Pérez González # 2, Anat Loewenstein 2, Dinah Zur 2 Affiliations expand PMID: 34373607 DOI:10.1038/s41433-021-01722-8

-Retina. 2019 Nov;39(11):2161-2166. doi:10.1097/IAE.0000000000002270. UNDERDIAGNOSED OPTIC DISK PIT MACULOPATHY: Spectral Domain Optical Coherence Tomography Features For Accurate

Diagnosis Matias Iglicki¹, Catharina Busch², Anat Loewenstein^{3 4 5} Adrian T Fung^{6 7 8}, Alessandro Invernizzi^{8 9}, Miriana Mariussi¹⁰, Romina Arias¹, Pierre-Henry Gabrielle¹¹, Zafer Cebeci¹², Mali Okada¹³, Jerzy Nawrocki^{14 15}, Zofia Michalewska^{14 15}, Michaela Goldstein^{3 4}, Adiel Barak³, Dinah Zur³

-EYE-2021 Apr;35(4):1111-1116. doi:10.1038/s41433-020-01309-9. Epub 2021 Jan 11. Longer-acting treatments for neovascular age-related macular degeneration-present and future

-Ophthalmology.2018 Feb;125(2):267-275. doi: 10.1016/j.ophtha.2017.08.031. Epub 2017 Sep 19. OCT Biomarkers as Functional Outcome Predictors in Diabetic Macular Edema Treated with Dexamethasone Implant

- Br J Ophthalmol 2020 Dec 7;bjophthalmol-2020-317422. doi: 10.1136/bjophthalmol-2020-317422. Online ahead of print. Central serous chorioretinopathy imaging biomarkers

-Eye .2022 Feb;36(2):273-277. doi: 10.1038/s41433-021-01722-8. Epub 2021 Aug 9. Next-generation anti-VEGF agents for diabetic macular oedema

-Retina2022 Mar 2. doi: 10.1097/IAE.0000000000003466. Online ahead of print. 'Target Sign' - A near infrared feature and multimodal imaging in a pluri-ethnic cohort with RDH5-related fundus albipunctatus. please add how, and how long takes for a retina specialist to think about OCT biomarker to start the proper treatment

-Ophthalmol Retina 2021 Nov;5(11):1097-1106. doi: 10.1016/j.oret.2021.01.013. Epub 2021 Feb 1. Detection of Diabetic Retinopathy from Ultra-Widefield Scanning Laser Ophthalmoscope Images: A Multicenter Deep Learning Analysis. Results could be misinterpreted add a short summary of the similarities in different devices where can be added the algorithms and also add different OCT modalities

-Eye .2022 Aug 29. doi: 10.1038/s41433-022-02222-z. Online ahead of print. Naïve subretinal haemorrhage due to neovascular age-related macular degeneration. pneumatic displacement, subretinal air, and tissue plasminogen activator: subretinal vs intravitreal aflibercept-the native study

Please apply correction for misspelling and

English grammar. This paper should be corrected by an english redactor	
Reviewer 2	
It is a well-written manuscript that analyzes the cost-effectiveness between SL and SML.	Thank you for your comment. No response needed.
It would be appropriate to discuss and compare other studies that evaluate costs and utilities regarding laser treatment and SML for DME.	Thank you for your comment. To the best of our knowledge, our HTA report (Lois N, Campbell C, Waugh N, Azuara-Blanco A, Maredza M, Mistry H, et al. DIAbetic Macular Oedema aNd Diode Subthreshold micropulse laser (DIAMONDS): A pragmatic, multicentre, allocation concealed, double-masked prospective, randomised, non-inferiority, clinical trial. Health Technology Assessment. 2022;26(50):1-86) was the only review that has addressed this. There are no other clinical studies that have evaluated the costs and utilities for SML for DME.
Page 15 line 14 ... potentially indefinitely, as needed... I would recommend avoiding stating that the treatment would be indefinite.	Thank you. We have removed these words from the sentence.
Reviewer 3	
The work seems complete and exhaustive, well written for methods, results and discussion, with great data presentation and analysis. The study could be eligible for publication.	Thank you for your comment. No response needed.
Reviewer 4	
The study topic of analyzing the costs difference between SML (subthreshold micropulse laser) and SL (standard laser) treatments for Diabetic Macular Edema is interesting and relevant, as both treatments are commonly used in clinical practice. The study builds upon the results of a well-conducted clinical trial (DIAMONDS) that compared the efficacy of SML, and SL treatments for DME. The study's focus on cost-effectiveness is especially important given the increasing pressure on healthcare systems to deliver high-quality care while containing costs. Before proceeding further, a few points need to be clarified to ensure that the study's results and conclusions are reliable and valid:	Thank you for your comment. No response needed.
Comment 1: The results of the analysis presented in the report are somewhat confusing and difficult to interpret. Furthermore, the confidence interval for the estimated cost-effectiveness ratios is relatively wide, which indicates a high degree of uncertainty around the results. However, the report suggests that the costs of the SML (subthreshold micropulse laser) and SL (standard laser) treatments are comparable. Similarly, the tables presented in the report suggest that SML may be more cost-effective than SL, despite the similar QALY (quality-adjusted life year) estimates. This finding may be counterintuitive and warrants further investigation to understand the potential reasons for this unexpected result.	We thank the reviewer for this comment, and we agree that the finding may be counterintuitive. We have revised the following text in the first paragraph of the discussion to help explain this better to the reader. "Costs and outcome data were collected prospectively. We found no significant differences in EQ-5D-5L and only a trivial non-significant difference of 0.008 in a calculation of QALYs. So, the verdict on whether one form of laser is better than the other shifts the focus onto the costs. This analysis found that costs were slightly higher (but not statistically significantly so) in the SL arm compared to the SML arm, due to more participants in the SL arm needing higher numbers (10 or more) of

	anti-VEGF rescue injections. In economic terms bringing both the costs and QALYs together in a ratio form, this meant the ICER for the base-case analysis indicated that SML is the dominant procedure, as average costs for this intervention were lower and average benefits were marginally higher than those for SL. However, caution should be taken when interpreting the results, given the wide confidence intervals around the mean costs (which ranges from cost saving to cost increasing). Taking this into consideration, costs of SML and SL treatment arms seem comparable.”
Comment 2: Table 2 in the report presents the cost estimates for the SML (subthreshold micropulse laser) and SL (standard laser) treatments, and it is important to note that the cost difference between the two treatments depends on several factors. These include whether the consultation is led by a consultant, the number of rescue treatments required, and the number of anti-VEGF injections administered. However, the results for the number of anti-VEGF injections should be viewed with caution, as they are heavily skewed by a small number of patients in the SL group who received more than 10 injections. The factor on consultant led / non-consultant led clinic is probably a random factor, and should be excluded in the analysis.	Thank you for your comment. The point about consultant led/non-consultant led clinic is a random factor and should be excluded. We do not agree that this should be excluded, because in the CRFs we had recorded who had undertaken the clinic (i.e. the grade of the professional) and therefore these clinics have been costed appropriately.
Furthermore, the published costs for anti-VEGF agents were used, which may not accurately reflect the actual costs incurred by the NHS. As a result, the reported cost difference between the SML and SL treatments may be exaggerated.	Thank you for your comment – this point has already been noted in the discussion (see paragraph 4).
Reviewer 5	
Overall, the analysis by Mistry et al. which performed a cost-effectiveness analysis from the DIAMONDS trial fulfills most of the core recommendations for conducting economic analyses alongside a clinical trial. However, before publication can be recommended, some aspects have to be clarified and improved.	Thank you for your comment. No response needed.
• Page 5, line 5-8: the authors state that their economic study is based on data obtained from an appropriately powered pragmatic randomised clinical trial. This might be true for the clinical endpoint (BCVA) but is not for the issue of cost-effectiveness. Whereas it is common to base a trial-based cost-effectiveness study on the sample size of the trial, ideally, researchers should set up a formal hypothesis for the cost-effectiveness analysis (to avoid underpowerment, see Ramsey et al. Value in Health 2015). This should be stated as a limitation, particularly because the primary study endpoint was not used here.	Thank you for your comment. We have added some text in the discussion to allude to the fact that BCVA is not the primary outcome in the cost-effectiveness analysis.

 • Page 9, line 25-50: the authors state that the EQ-5D was used for generating QALYs. In the same paragraph the NEI-VFQ-25 and the VisQoL are mentioned, with the latter being used for a time-trade off exercise. Please provide more information about the necessity to add these two instruments for assessing HRQoL. QALYs were derived from the EQ-5D and I would have expected information about the tariffs used for valuing these QALYs obtained from the study. I do not understand how the NEI-VFQ-25 and the VisQoL were incorporated in the analysis. Did the tables A2 and A3 play any role for the analysis? 	We have added further information into the text to say what tariff we used to estimate utility scores for the EQ-5D responses. We have also separated the text into two paragraphs so this is now clearer to the reader that the VFQ-25 and VisQoL was only presented as summary scores and not used in the analysis for utility/QALY calculation.
 • Bootstrap: a graphical illustration would be useful. 	We thank the reviewer and we have now included Figure 1 which shows graphical illustration of the bootstrap results. We have written this at the end of the results section. “Figure 1 shows the joint distributions of costs and outcomes.”
 • Discussion: overall, this section is disappointing. Several aspects potentially affecting the internal and external validity of the analysis should be discussed more extensively (e.g., performing an ITT-analysis for the economic evaluation but a PP-analysis for the trial, or the impact of calculating a protocol-driven resource use). 	We thank the reviewer for this comment. In our HEAP we stated that we would do an ITT analysis as this is what is the usual approach for analysis for economic evaluations. In our HTA report as part of a sensitivity analysis we also conducted a PP analysis in line with the SAP and we found that the results were very similar: SML was still dominant (incremental costs and QALYs were -£374 and 0.0077, respectively) and the probability of SML being cost-effectiveness at the £20k/QALY was 0.773, hence we didn't think of including this in the paper. We have now added a few sentences at the end of the first paragraph into the discussion to say that the PP analysis results were similar to the ITT results. “We also conducted a per-protocol analysis as part of a sensitivity analysis as DIAMONDS was a non-inferiority trial. The results were in line with the intention-to-treat analysis were SML remained the dominant option and the probability of of SML being cost-effectiveness at the £20k/QALY threshold was 0.773.”
 • Supplemental material: the authors added the CHEERS-checklist for demonstrating reporting quality which should be mentioned in the method section (with reference). In addition, it would be useful to consider the work of Ramsey et al. for assuring the methodological quality (Ramsey et al. 2005 and 2015). 	We have added a reference to the CHEERS checklist in the text.
 • Supplemental material (CHEERS-checklist): many of the items are responded with N/A. At least for item 20 and 24 (uncertainty) this should be updated because uncertainty was analyzed (e.g., bootstrap). For others (e.g., item 18) one 	Thank you for your comment. We have updated the CHEERS checklist.

could argue that the items were not addressed (but could have).	
• It is conspicuous that only a small number of references was used for the study and no additional references in the discussion to put study results in a broader context. Please revise.	We thank the reviewer for this comment. We believe that we have now strengthened the discussion by adding some more text into all of the paragraphs in the discussion section, backed up by additional references.

VERSION 2 – REVIEW

REVIEWER	Ho, Mary The Chinese University of Hong Kong
REVIEW RETURNED	11-Sep-2023

GENERAL COMMENTS	The revised manuscript has effectively addressed all previously raised questions and provided a clear rationale for the chosen approach. Significant improvements have been made in the presentation of cost-effectiveness, making it easier for readers to understand the scale of the problems and the costs associated with either laser treatments.
------------------	---

REVIEWER	Müller, Dirk University Hospital Cologne, Institute of Health Economics and Clinical Epidemiology
REVIEW RETURNED	04-Sep-2023

GENERAL COMMENTS	Most comments were responded sufficiently. Before the paper can be recommended three additional points should be clarified.  1. Introduction: This part focused on the clinical situation with regard to the DIAMOND-trial but completely lacks to provide information about the economic burden of DMO (and the requirement for an economic analysis). 2. The authors conclude to prefer SML to SL and justify this with the fact that SML does not burn the retina (unlike SL). This is rather surprising with regard to the bootstrap-result which indicate a considerable probability for SML being cost saving (I would expect higher than those presented for being cost-effective at a threshold of £20,000. If this is true, the conclusion should be revised. With regard to the parameter burn: was this analysed in the DIAMOND-trial? If so, details should be presented (in the revised version of the manuscript the argument is only presented in the discussion without details). Please clarify. 3. In addition, a few changes could be made with regard to language style (e.g., „In economic terms bringing both the costs and QALYs together in a ratio form...“) CHEERS-checklist: changes for items 20 and 24 are not provided (only deletion)
------------------	---

VERSION 2 – AUTHOR RESPONSE

Reviewers comments	Our response
Reviewer 5	
Most comments were responded sufficiently. Before the paper can be recommended three additional points should be clarified.	Thank you for your comment. No response needed.
1. Introduction: This part focused on the clinical situation with regard to the DIAMOND-trial but completely lacks to provide information about the economic burden of DMO (and the requirement for an economic analysis).	Thank you for your comment, we have added in some further text into the introduction section. In the first paragraph with regard to the economic burden of DMO and in the second paragraph with regard to the lack of cost-effectiveness studies conducted comparing SML compared with SL.
2. The authors conclude to prefer SML to SL and justify this with the fact that SML does not burn the retina (unlike SL). This is rather surprising with regard to the bootstrap-result which indicate a considerable probability for SML being cost saving (I would expect higher than those presented for being cost-effective at a threshold of £20,000. If this is true, the conclusion should be revised. With regard to the parameter burn: was this analysed in the DIAMOND-trial? If so, details should be presented (in the revised version of the manuscript the argument is only presented in the discussion without details). Please clarify.	Thank you for your comment. Foveal burn were considered as an adverse event in DIAMONDS and this event was recorded prospectively throughout the trial. However, neither arm experienced this adverse event (see HTA monograph, p35). Hence, 'foveal burns' did not have any impact on the cost-effectiveness results. Burns outside the fovea were not investigated in DIAMONDS. There was extensive evidence using numerous functional and structural studies prior to DIAMONDS demonstrating that standard laser produces retinal burns, but SML does not, so we did not look at this specifically in the trial. We have, thus, not changed the manuscript further as a result of this comment from the Reviewer.
3. In addition, a few changes could be made with regard to language style (e.g., „In economic terms bringing both the costs and QALYs together in a ratio form...“)	Thank you for the comment, we have now revised this sentence and replaced the words 'In economic terms bringing' with 'Reporting' to make the sentence clearer.
CHEERS-checklist: changes for items 20 and 24 are not provided (only deletion)	We apologise for this. We had put the new page numbers in as a comment as it did not allow us to edit the file. We have now uploaded a new version with all the page numbers visible.
Reviewer 4	
The revised manuscript has effectively addressed all previously raised questions and provided a clear rationale for the chosen approach. Significant improvements have been made in the presentation of cost-effectiveness, making it easier for readers to understand the scale of the problems and the costs associated with either laser treatments.	Thank you for your comment. No response needed.

VERSION 3 – REVIEW

REVIEWER	Müller, Dirk University Hospital Cologne, Institute of Health Economics and Clinical Epidemiology
REVIEW RETURNED	28-Sep-2023

GENERAL COMMENTS	With regard to my second comment (no changes made), again I would recommend to more emphasize the saving potential of the intervention.
------------------	---

VERSION 3 – AUTHOR RESPONSE

Reviewers comments	Our response
Reviewer 5	
With regard to my second comment (no changes made), again I would recommend to more emphasize the saving potential of the intervention.	Thank you for your comment. We have now added in the suggested change at the end of the results section. “The graph also highlights that SML has the potential to be cost saving as the majority of the bootstrapped iterations lie in the bottom half of the cost-effectiveness plane.”